# Charge Transfer in Mixed-Phase TiO₂ Photoelectrodes for Perovskite Solar Cells

**Anna Nikolskaia [1],\*, Marina Vildanova [1], Sergey Kozlov [1], Nikolai Tsvetkov [1,2], Liudmila Larina [1,3] and Oleg Shevaleevskiy [1],\***

[1]   Solar Photovoltaic Laboratory, Emanuel Institute of Biochemical Physics, Russian Academy of Sciences, 119334 Moscow, Russia; mvildanova@sky.chph.ras.ru (M.V.); kozlovss@sky.chph.ras.ru (S.K.); nickts@kaist.ac.kr (N.T.); larina@kaist.ac.kr (L.L.)

[2]   Department of Energy Environmental Water and Sustainability, Korea Advanced Institute of Science and Technology, Daejeon 305-701, Korea

[3]   Department of Chemical Engineering and Applied Chemistry, Chungnam National University, Daejeon 34134, Korea

\*   Correspondence: anikolskaia@sky.chph.ras.ru (A.N.); shevale@sky.chph.ras.ru (O.S.)

**Abstract:** In mesoscopic perovskite solar cells (PSCs) the recombination processes within the TiO₂ photoelectrode and at the TiO₂/perovskite interface limit power conversion efficiency. To overcome this challenge, we investigated the effect of TiO₂ phase composition on the electronic structure of TiO₂ photoelectrodes, as well as on PSCs performance. For this, a set of PSCs based on TiO₂ thin films with different content of anatase and rutile particles was fabricated under ambient conditions. X-ray diffraction, optical spectroscopy and scanning electron microscopy were used to study the structural, morphological and optical characteristics of TiO₂ powders and TiO₂-based thin films. X-ray photoelectron spectroscopy (XPS) analysis of anatase revealed a cliff conduction band alignment of 0.2 eV with respect to the rutile. Energy band alignment at the anatase/rutile/perovskite interfaces deduced from the XPS data provides the possibility for interparticle electron transport from the rutile to anatase phase and the efficient blocking of electron recombination at the TiO₂/perovskite interface, leading to efficient electron-hole separation in PSCs based on mixed-phase TiO₂ photoelectrodes. PSCs based on TiO₂ layers with 60/40 anatase/rutile ratio were characterized by optimized charge extraction and low level of recombination at the perovskite/TiO₂ interface and showed the best energy conversion efficiency of 13.4% among the studied PSCs. Obtained results provide a simple and effective approach towards the development of the next generation high efficiency PSCs.

**Keywords:** perovskite solar cells; titanium dioxide; rutile; anatase; optoelectronic structure

## 1. Introduction

To date the key challenge addressed in solar cell technology is to provide high efficiency of devices while keeping low production costs and environmental safety [1,2]. Perovskite solar cells (PSCs), regarded as the third-generation photovoltaic technology, satisfied the aforementioned requirements and could be considered as a reasonable alternative to the well-developed crystalline silicon and thin film solar cell technologies [3,4]. Low cost fabrication processes based on solution-processed materials and high power conversion efficiency (PCE) create a platform for the rapid development of PSCs. Furthermore, the advantages of PSCs over crystalline silicon and thin-film solar cells in specific applications (building-integrated photovoltaics, portable electronics, IoT devices etc.) accelerate PSCs technology development.

Today the PCE of PSCs reached 25% value on a laboratory scale in a glove-box, which is currently competitive to the efficiency of 26–27% for crystalline silicon solar cells [5,6] and 22.9%

efficiency for CIGS solar cells [7]. PSCs are based on compounds with the chemical formula $ABX_3$, where $A = CH_3NH_3^+$, $HC(NH_2)_2^+$, $Cs^+$; $B = Pb^{2+}$, $Sn^{2+}$; $X = I^-$, $Br^-$, $Cl^-$. These compounds are excellent light absorbers and charge transporting materials with ambipolar properties [8–10]. The perovskite layer is usually sandwiched between the electron transport layer (ETL) and hole transport material [11]. The effect of the ETL on the photovoltaic (PV) parameters of the PSCs is determined by its structural and charge transport characteristics. One of the limitations of solar cell performance is related to the efficiency of the electron injection from perovskite to ETL and the efficiency of photo-injected electron collection [12,13]. Thus, optimization of ETL charge transport characteristics is required in order to reduce interface recombination and increase PSCs performance.

A nanostructured mesoscopic titanium dioxide ($TiO_2$) layer is usually applied as an ETL for PSC photoelectrodes. The main advantages of wide-bandgap semiconductor $TiO_2$ over other metal oxides include its excellent thermal stability, non-toxicity and passivity to chemicals or electrolytes used in solar cells [14–16]. The most abundant structural phases of $TiO_2$ are anatase and rutile. Anatase sphere-like nanoparticles are characterized by higher electron mobility and a lower charge recombination rate than the rod-like structure of the rutile phase. At the same time, rutile is thermodynamically more stable with better light reflecting properties compared to anatase [17].

Previously several studies were performed on the investigation of the dependence of dye-sensitized solar cells (DSCs) performance on the anatase/rutile phase ratio in the nanostructured mesoscopic photoelectrodes [18–20]. It was found that the addition of the rutile to anatase in the range of 10–15% significantly improved light harvesting and the overall power conversion efficiency of DSCs [18]. Yun et al. found some controversial results: incorporation of rutile particles into the $TiO_2$ anatase layer reduced the specific surface area leading to decreased dye adsorption. However, the best photoelectrical parameters were achieved for DSC based on the $TiO_2$ layer with the anatase/rutile ratio of 84/16% [19]. These results suggest the existence of another positive impact of rutile addition on the PCE that dominates the effect of the specific surface decrease [20]. Similar phenomena described in references [21,22] also confirms this suggestion.

Later it was proven that the observed results were associated with the synergistic effect, manifested in the suppression of recombination processes on the surface of $TiO_2$ particles with a specified composition of anatase and rutile phases [23,24]. This effect is the consequence of differences in morphology (sphere vs. rod-like), charge transfer dynamics and electronic structure between anatase and rutile. The band alignment between anatase and rutile can provide an additional driving force for charge transport, and the values of valence and conduction band discontinuities are still a subject for discussion [25,26]. Scanlon et al. [26] characterized the electronic structure of the anatase/rutile interface as spike type alignments for both the valence and conduction bands (the energetic levels of rutile are above energetic levels of anatase) employing X-ray photoelectron spectroscopy. Consequently, the photogenerated electrons are transferred from rutile to anatase and holes are moved from anatase to rutile. The difference in the electronic structure of the rutile and anatase phases could be the key factor of the phenomena observed in mixed-phase $TiO_2$ photoelectrodes [19,21,22].

Given the aforementioned, it is essential to study the influence of the $TiO_2$ phase composition on PV characteristics of PSCs when designing a solar cell. To the best of our knowledge, there is currently only limited data in the literature regarding the correlation between PV parameters of the PSCs and the structural properties of the mixed-phase $TiO_2$ photoelectrodes, which leaves some open questions in this respect. The known data are related to the performance of PSCs based on the $TiO_2$ photoelectrodes containing either pure anatase phase, or pure rutile phase. Currently all state-of-the-art mesoscopic PSCs are based on anatase ETLs. On the other hand, some studies show that depending on the particle synthesis method and perovskite deposition technique it is possible to obtain rutile-based PSCs with efficiencies higher than anatase-based ones [27]. Therefore, additional studies on the performance and charge transfer dynamics in PSCs based on anatase, rutile and mixed-phase $TiO_2$ ETLs are necessary.

In this study we investigated the influence of the $TiO_2$ phase composition on the electronic structure of photoelectrode and, consequently, on the charge transfer processes in PSCs. The PV

performance of PSCs with a structure FTO/compact layer/TiO$_2$/CH$_3$NH$_3$PbI$_3$/Spiro-MeOTAD/Au was studied. Scanning electron microscopy (SEM), X-ray photoelectron spectroscopy (XPS), current density-voltage (J-V), external quantum efficiency (EQE) and electrochemical impedance spectroscopy (EIS) measurements were provided. This is the first systematic study on the effects of the optoelectronic structures in mixed-phase TiO$_2$ on PV parameters of PSCs. The obtained results provide a novel effective approach towards the development of the next generation of high-efficient PSCs.

## 2. Materials and Methods

### 2.1. Materials

FTO-coated glasses were purchased from Solaronix (Aubonne, Switzerland). Commercial TiO$_2$ powder (Aeroxide P25, Acros Organics, Geel, Belgium) was used for the synthesis of rutile particles. Triton X-100 was purchased from AppliChem (Darmstadt, Germany). Anatase nanoparticles (<25 nm, 99.7%), titanium diisopropoxide dis(acetylacetonate) (TAA, 75 wt% in isopropanol), acetone, diethyl ether, ethanol, acetic acid, ethylcellulose, terpineol, 1-butanol, Spiro-MeOTAD, 4-tert-butylpiridine, lithium bis(trifluoromethanesulfonyl)imide (LiTFSI), chlorobenzene were purchased from Sigma-Aldrich (St. Louis, MO, USA). Methylammonium iodide (CH$_3$NH$_3$I) and lead iodide (PbI$_2$) were purchased from TCI Chemicals (Tokyo, Japan). Dimethyl sulfoxide (DMSO) and N,N-dimethylformamide (DMF) were purchased from Panreac (Barcelona, Spain). Acetonitrile was purchased from Macron Fine Chemicals (Radnor, PA, USA). All chemicals were ACS-grade reagents.

### 2.2. Synthesis of Nanostructured TiO$_2$ Films

The Solaronix glasses of $2 \times 2$ cm size, covered by fluorine-doped tin oxide conductive layer (FTO), were pre-cleaned in ultrasonic bath using Triton X-100, ethanol and acetone with subsequent drying in argon flow and were used as substrates. To prevent electrical contact between perovskite material and FTO, the solution of 0.15 M TAA in 1-butanol was spin-coated on glass substrate at 2000 rpm for 1 min, followed by heating at 130 °C for 5 min [28].

Pure rutile particles were synthesized by calcination of commercial TiO$_2$ powder at 800 °C during 2 h. A set of TiO$_2$ powder samples with varied anatase/rutile ratios (100/0, 80/20, 60/40, 40/60, 20/80 and 0/100%) was prepared by mechanical treatment in a ball mill for 3 h and was used for fabrication of TiO$_2$ pastes following the known procedure [29]. The samples were denoted as A/R 80/20, A/R 60/40, A/R 40/60, A/R 20/80. Pure anatase and rutile powders were labeled as A and R, respectively. To produce TiO$_2$ pastes the powders were mixed with acetic acid, ethanol, anhydrous terpineol and two types of ethyl cellulose (5–15 mPa·s and 50–70 mPa·s, 1:1 *w/w*). Prepared solutions were sonicated several times and then ethanol was evaporated at 80 °C. TiO$_2$ pastes were diluted by ethanol at the 1:10 mass ratio and were spin-coated (2000 rpm, 1 min) on the glass substrates. The fabricated mesoscopic TiO$_2$ layers were annealed at 500 °C during 30 min, followed by treatment in 20 mM aqueous TiCl$_4$ solution for 10 min at 90 °C and re-annealing at 500 °C for 30 min.

### 2.3. Fabrication of PSCs

The fabrication of PSCs was carried out under ambient conditions (humidity ~50–60%), following the known technology [28]. The solution of perovskite CH$_3$NH$_3$PbI$_3$ was prepared by mixing 461 mg PbI$_2$, 159 mg CH$_3$NH$_3$I and 71 μL DMSO in 635 μL of DMF (a molar ratio of 1:1:1). This solution was spin-coated on the TiO$_2$ photoelectrode at 4000 rpm for 25 s by a one-step deposition method using diethyl ether as antisolvent. Obtained perovskite layers were dried at 100 °C for 10 min. At the next step, the hole-transporting material Spiro-MeOTAD was deposited by spin-coating at 2000 rpm for 30 s. The following solution was used: 72.3 mg Spiro-MeOTAD, 28.8 μL 4-tert-butylpiridine and 17.5 μL Li-TSFI solution (520 mg Li-TSFI in 1 mL of acetonitrile) in 1 ml of chlorobenzene. The process of PSC fabrication was completed by the deposition of the Au electrodes with a thickness of 50 nm using thermal evaporation.

*2.4. Characterization*

The structure and the composition of $TiO_2$ particles were investigated by X-ray diffraction (XRD) measurements using a DRON-3M X-ray diffractometer with Cu K$\alpha$ radiation ($\lambda$ = 1.5405 Å) as the X-ray source. Scans were taken in the 2$\theta$ range of 20–75° with the 0.1° (2$\theta$) scan step and counting time per data point of 5 s. The content of anatase and rutile in all $TiO_2$ samples was calculated according to [30]. Diffuse reflectance spectra of $TiO_2$ powders with mixed-phase were recorded using Shimadzu UV−3600 spectrophotometer with an ISR-3100 integrating sphere (Kyoto, Japan) in the wavelength range of 300–1200 nm. The equipment was calibrated using barium sulfate powder as a standard. The morphology of the $TiO_2$ photoelectrodes was investigated using a dual-beam scanning electron microscope (SEM) Helios NanoLab 660 (FEI, Hillsboro, OR, USA).

The XPS measurements were performed with a K-Alpha spectrometer (Thermo Scientific, Waltham, MA, USA) equipped with monochromatic Al K$\alpha$ source (1486.6 eV). Calibration of the spectra was performed by referring to the C 1s peak (C-C bond) at 285.0 eV. The energy of the valence band maximum (VBM) positions of the samples was determined from the emission spectra through linear extrapolation of the steep leading edge of the highest valence band peak to the baseline.

The measurements of PV parameters for PSCs were carried out under standard AM1.5G illumination conditions (1000 W/m$^2$) using Abet 10,500 Solar Simulator (Abet Technologies, Milford, CT, USA). The J–V characteristics were measured using the SCS-4200 Semiconductor Characterization System (Keithley, Beaverton, OR, USA). Average values of PV parameters were obtained for a series of 30 samples for each PSC type. The EQE spectra were recorded using QEX10 Solar Cell Quantum Efficiency Measurement System (PV Measurements, Point Roberts, WA, USA) by scanning the wavelength of the incident monochromatic light in the range of 300–900 nm and measuring the current density at 10 nm intervals. The EIS measurements were provided using P-45X potentiostat (Elins, Zelenograd, Russia) equipped with the frequency response analyzer. The EIS measurements were performed under AM1.5G simulated illumination in the frequency range of 500 kHz–1 Hz with the amplitude of the modulated voltage of 20 mV. Experimental EIS data were fitted using ZView software (Scribner Associates, Southern Pines, NC, USA).

## 3. Results and Discussion

The purity of commercial anatase (A) and synthesized rutile (R) phases and the high quality of their mixing were confirmed by XRD analysis and optical spectroscopy. XRD patterns for A and R powders as well as for synthesized powders with mixed structural phases (A/R) are presented in Figure 1. The content of A and R phases in all samples under study was determined on the basis of the intensity ratio of the main anatase (101) and rutile (110) peaks. Results of calculations are given in Table 1.

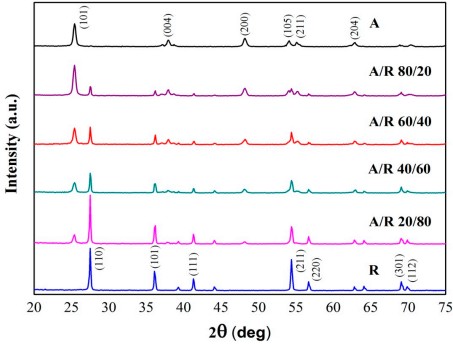

**Figure 1.** XRD patterns of $TiO_2$ powders with different anatase/rutile (A/R) phase composition.

**Table 1.** The content of anatase (A) and rutile (R) phases in $TiO_2$ powder samples calculated from XRD patterns.

| Sample | Anatase Content (%) | Rutile Content (%) |
|--------|---------------------|--------------------|
| A | 100 | 0 |
| A/R 80/20 | 81.5 | 18.5 |
| A/R 60/40 | 60.1 | 39.9 |
| A/R 40/60 | 41.6 | 58.4 |
| A/R 20/80 | 21.7 | 78.3 |
| R | 0 | 100 |

Diffuse reflectance spectra for $TiO_2$ powders with different A/R phase composition are shown in Figure 2a. As can be seen, pure rutile and mixed-phase powders show increased reflectivity in the visible region as compared to pure anatase powder, presumably due to larger $TiO_2$ particle size. Moreover, the addition of rutile to anatase leads to a shift in the absorption edge to higher wavelengths, indicating the change in the $E_g$.

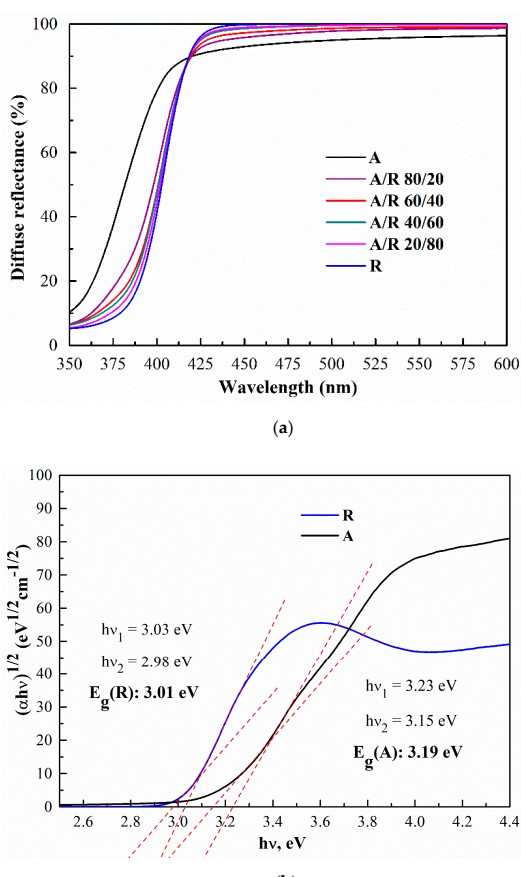

(**a**)

(**b**)

**Figure 2.** (**a**) Diffuse reflectance spectra for $TiO_2$ powders with different A/R phase composition and (**b**) $E_g$ values for anatase (A) and rutile (R) powders extracted from $(\alpha h \nu)^{1/2}$ vs. photon energy plots.

Given that $TiO_2$ nanoparticles are a highly light scattering material, the $E_g$ for these samples were evaluated by applying the Kubelka–Munk method [31], which is based on the following equation:

$$F(R) = \frac{(1-R)^2}{2R}$$

where $R$ is diffuse reflectance, $F(R)$ is the Kubelka–Munk function, which is proportional to the absorption coefficient ($\alpha$). For an indirect semiconductor such as $TiO_2$, the expected variation of $hv{\cdot}F(R)$ with the photon energy $hv$ near the absorption edge can be expressed as follows:

$$hv{\cdot}F(R) \; = \; B\big(hv - E_g\big)^2$$

where $B$ is the absorption constant for the indirect transition, $h$ is Plank's constant, $v$ is the frequency [32]. The analysis of the diffuse reflectance spectra in terms of indirect optical absorption are shown in Figure 2b. For anatase powder the plot of $(\alpha hv)^{1/2}$ versus photon energy gives two straight line segments with two intercepts on the energy axis, $hv_1$ and $hv_2$. The $E_g$ value could be estimated as $\frac{1}{2}(hv_1 + hv_2)$, yielding the values of 3.19 eV for anatase and 3.01 for rutile. The obtained $E_g$ values are in agreement with the data reported in the literature [26,32].

Photoelectrodes for PSCs were fabricated using all prepared $TiO_2$ powder samples. SEM measurements of $TiO_2$ mesoscopic layers deposited on the FTO glass substrate reveal that the average size of anatase particles is about 40 nm, whereas the average size of rutile particles is 200 nm (Figure 3). The particle size distributions for anatase and rutile are given in Figure S1. SEM images of $TiO_2$ mesoscopic layers showed that anatase ETL possess uniform and smooth morphology. Rutile ETL showed less smooth morphology as compared to anatase, with visible voids and non-uniformity of the $TiO_2$ layer, predominantly due to large particle size (~200 nm). SEM images of $TiO_2$ thin films prepared with 20% (A/R 80/20) and 40% (A/R 60/40) of rutile revealed that rutile particles formed isolated clusters inside the anatase phase. A similar trend was observed for the anatase phase in $TiO_2$ films prepared with 20% (A/R 20/80) and 40% (A/R 40/60) of anatase. This suggests that nanoparticles with the same crystalline phase have a tendency to agglomerate inside the mixed-phase $TiO_2$ ETLs.

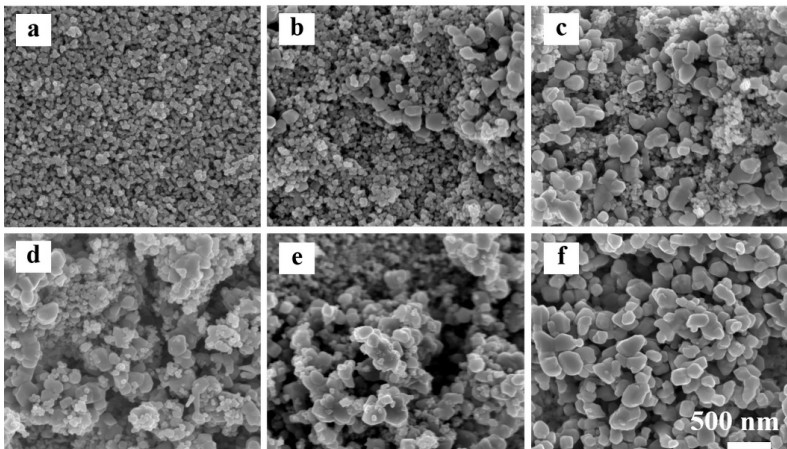

**Figure 3.** SEM images of $TiO_2$ mesoscopic layers with different A/R ratios (**a**—A, **b**—A/R 80/20 R, **c**—A/R 60/40, **d**—A/R 40/60, **e**—A/R 20/80, **f**—R). The scale bar is the same for all images.

Photoelectrodes based on mesoscopic $TiO_2$ layers with a mixed phase structure were used for PSCs fabrication under ambient conditions with high humidity of around 50–60%. J-V curves obtained under AM1.5G illumination (1000 W/m$^2$) are shown in Figure 4a together with the zoomed-in J-V plots in the vicinity of open-circuit voltage $V_{OC}$ (Figure 4b). Corresponding EQE spectra are shown in Figure S2. The average values of PV parameters for all PSCs types are given in Table 2. Histograms of PV parameters for the PSCs based on anatase, rutile and mixed-phase $TiO_2$ ETLs for a series of 30 samples are given in Figure S3. The maximum EQE values in the wavelength range of 350–750 nm (Figure S2) are gradually decreased from 80% to 60% with an increase of rutile content. According to Figure 4 and Table 2, the highest $V_{OC}$ value was obtained for PSC with ETL based on pure rutile. Solar cells with pure anatase ETLs or with A/R 80/20 and A/R 60/40 layers demonstrated the highest values of short-circuit current density ($J_{SC}$). The maximum average PCE of 13.4% was achieved for PSC based

on $TiO_2$ ETL with a 60/40 A/R ratio. PSCs with predominant rutile content (A/R 40/60, A/R 20/80) are inferior in their performance compared to other samples. It might be due to the large size of rutile particles in comparison to the anatase particles and corresponding non-uniformity of ETLs (Figure 3).

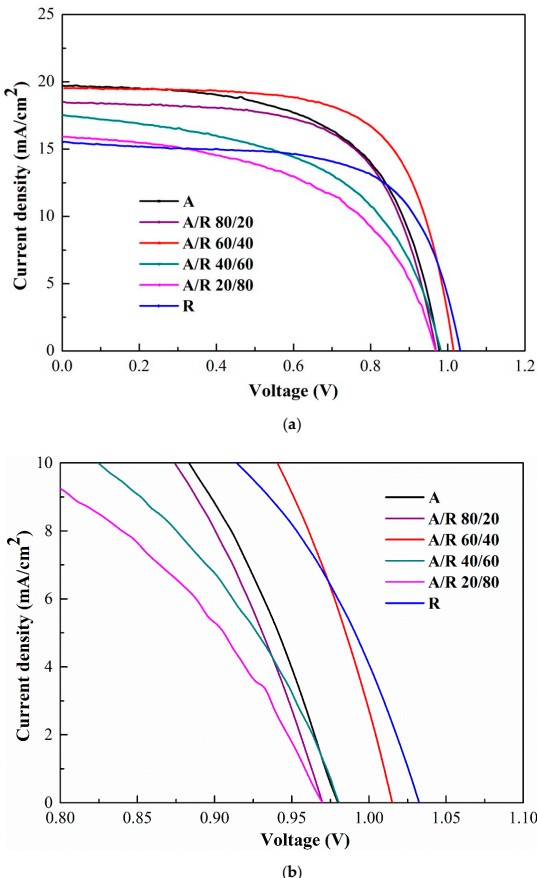

**Figure 4.** (**a**) J-V characteristics for PSCs based on $TiO_2$ photoelectrodes with different A/R ratios obtained under standard illumination conditions (1000 W/m², AM1.5G) and (**b**) zoomed-in J-V plots in the vicinity of $V_{OC}$.

**Table 2.** The average PV parameters of PSCs based on $TiO_2$ photoelectrodes with varied A/R ratios (mean ± SD).

| Sample | $J_{SC}$ (mA/cm²) | $V_{OC}$ (V) | FF | PCE (%) |
|---|---|---|---|---|
| A | 19.6 ± 0.18 | 0.98 ± 0.01 | 0.60 ± 0.013 | 11.6 ± 0.68 |
| A/R 80/20 | 18.5 ± 0.12 | 0.97 ± 0.01 | 0.64 ± 0.008 | 11.4 ± 0.9 |
| A/R 60/40 | 19.5 ± 0.14 | 1.02 ± 0.01 | 0.67 ± 0.007 | 13.4 ± 0.70 |
| A/R 40/60 | 17.5 ± 0.13 | 0.98 ± 0.01 | 0.53 ± 0.006 | 9.2 ± 0.72 |
| A/R 20/80 | 15.9 ± 0.15 | 0.97 ± 0.01 | 0.53 ± 0.007 | 8.2 ± 0.68 |
| R | 15.5 ± 0.17 | 1.03 ± 0.01 | 0.66 ± 0.007 | 10.5 ± 0.66 |

Observed data could be explained by the synergistic effect related to the interaction between anatase and rutile particles within the $TiO_2$ mesoscopic layer. A similar effect previously reported for DSCs was attributed to the efficient interparticle electron transport from rutile to anatase, leading to increased current densities and lower recombination at low (15%) rutile content [19,21]. XPS and EIS measurements were carried out to elucidate the effect of PCE increase upon the addition of 40% of rutile in fabricated PSCs.

The XPS analysis was performed to compare the surface electronic structure of the pure anatase phase, pure rutile phase and mixed-phase $TiO_2$ mesoscopic layers deposited on FTO glass. The XPS

spectra are shown in Figure 5a. The binding energy scale is referred to the Fermi level. Variation in the intensity of photoelectron emission reflects a different surface electronic structure of the samples under study. The spectrum structure in the energy range of 0–10 eV is the valence band structure and reflects the valence band density of states of the anatase and rutile phases. The valence band edges of samples are predominated by O 2p emission. The higher electron yield from the rutile phase than from the anatase phase is visible in the spectra.

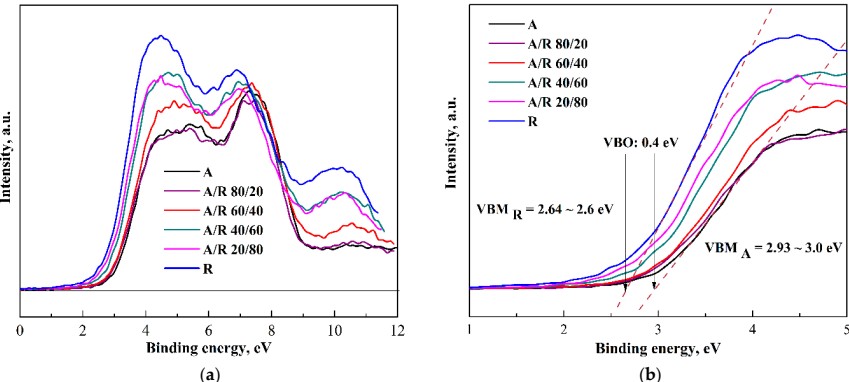

**Figure 5.** The XPS spectra taken from $TiO_2$ mesoscopic layers with different A/R phase ratios: (**a**) in the energy range of valence band, (**b**) shown on an expanded scale in the binding energy range of valence band. The binding energy scale is referred to the Fermi energy level.

A difference in the energy of the valence band edges is visible for the samples based on the mixed-phase $TiO_2$ mesoscopic layers. The same XPS spectra are shown in Figure 5b on an expanded scale in the valence band energy range. The VBM values for all samples obtained from XPS spectra are given in Table 3. It is estimated to be 3.0 eV and 2.6 eV for pure anatase and pure rutile phases, respectively. Taking into account the experimental uncertainty of 0.1 eV, the estimated VBM position for the anatase is in good agreement with VBM value reported by Liu et al. [33]. The determined VBM positions yield valence band offset (VBO) of 0.4 eV at the A/R interface. The identical result 0.39 eV was obtained from computational analysis [26]. Given that the bulk bandgap energy of the anatase and rutile were found to be of 3.2 eV and 3.0 eV, the estimated VBM positions place the Fermi energy level near the top of the fundamental gaps of the $TiO_2$, indicating high *n* doping for both phases.

**Table 3.** The VBM values (in eV) obtained for mesoscopic layers based on $TiO_2$ photoelectrodes with varied A/R phase ratio.

| Sample | A | A/R 80/20 | A/R 60/40 | A/R 40/60 | A/R 20/80 | R |
|---|---|---|---|---|---|---|
| VBM (eV) | 2.97 | 2.80 | 2.84 | 2.74 | 2.69 | 2.63 |

The conduction band discontinuity at the A/R interface was calculated using the energy bandgap values obtained from the optical analysis shown above in Figure 2b. It can be seen that the conduction band maximum (CBM) of rutile is located higher than the CBM of anatase. The conduction band offset (CBO) is 0.2 eV. Thus, the determined experimentally VBO at the A/R interface yields a cliff conduction band alignment of 0.2 eV with respect to the rutile. It can be assumed that the electron transport in $TiO_2$ photoelectrodes with mixed-phase composition is carried out from rutile particles to anatase particles. Electron migration from rutile to anatase was previously confirmed in the mixed-phase $TiO_2$ systems by transient infrared absorption-excitation spectroscopy [34]. The obtained results are in line with the theoretical predictions of both the Madelung-potential-based argument model and the defect model [26]. The calculated difference in carrier energies between the two materials corresponds to the experimentally determined energy band alignment in our study. Indeed, Scanlon et al. reported a 0.24 eV downward shift of the anatase conduction band as compared to that of rutile, and 0.39 eV

upward valence band shift of rutile compared to VBM of anatase. Thus, our results are consistent with the literature [26].

Figure 6 shows the scheme of the energy band alignments at the anatase/rutile/perovskite interfaces constructed on the basis of experimental VBM positions and bandgap values. Perovskite $CH_3NH_3PbI_3$ electronic structure was determined in our previous work [35].

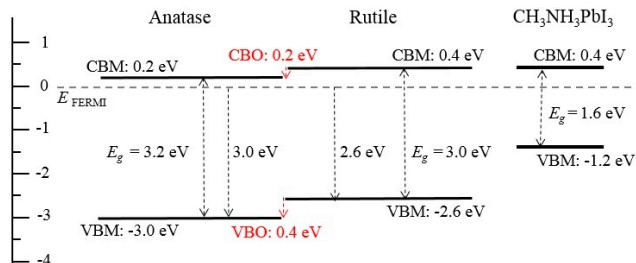

**Figure 6.** Energy band diagram at the anatase/rutile/perovskite interfaces constructed based on XPS and optical data. The binding energy scale is referred to as the Fermi energy level.

The EIS measurements of PSCs based on $TiO_2$ photoelectrodes with pure and mixed-phase composition were carried out. Figure 7 shows the Nyquist spectra of fabricated PSCs with different A/R ratios under 1 sun illumination and zero DC bias. The spectra exhibit a semicircle at the high frequency region and an incomplete arc at the low frequency region. The former corresponds to the charge transport phenomena in the PSC device, while the latter can be ascribed to the charge accumulation and recombination at the perovskite/$TiO_2$ interface [36,37]. Experimental Nyquist spectra were fitted using the equivalent circuit shown in Figure 7. The equivalent circuit consists of series resistance [$R_S$] accounting for the ohmic contribution of electric contacts and wires, charge transfer resistance [$R_{ct}$], and recombination resistance [$R_{rec}$]. Two constant phase elements (CPE) represent geometrical or bulk capacitance [$C_g$] associated with dielectric properties of the perovskite layer and capacitance attributed to charge accumulation at the perovskite/$TiO_2$ interface [$C_s$].

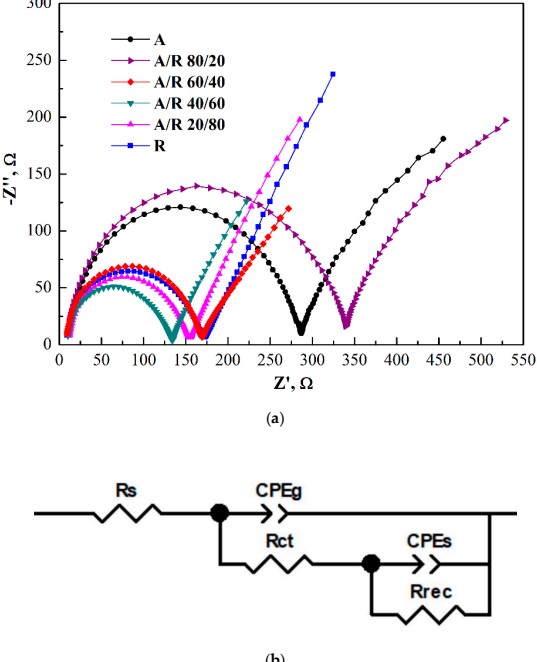

**Figure 7.** (**a**) Nyquist spectra for fabricated PSCs with different A/R ratios under 1 sun illumination and (**b**) the equivalent circuit used to fit the experimental Nyquist spectra.

Nyquist spectra indicate that fabricated PSCs fall into two distinctive groups. First, PSCs based on pure rutile and $TiO_2$ with 20/80, 40/60 and 60/40 A/R ratios; second, PSCs based on pure anatase and on $TiO_2$ with A/R ratio of 80/20. PSCs from the first group showed smaller high-frequency semicircle arc as compared to PSCs from the second one. Moreover, higher characteristic frequencies corresponding to charge transport across the device were observed in the Bode plots (Figure S4) for the former group. Obtained results suggest a better charge transport at the perovskite/$TiO_2$ interface for PSC devices related to the first group.

The results of modeling the Nyquist spectra for PSCs based on mesoscopic $TiO_2$ layers with different A/R ratios are listed in Table 4. PSCs based on $TiO_2$ layers with high rutile content (R, A/R 20/80, 40/60, 60/40) showed lower $R_{ct}$ values compared to anatase-based PSCs (A and A/R 80/20), confirming more efficient electron transfer across the perovskite/$TiO_2$ interface for the former. Rutile-based PSC showed the highest $R_{rec}$ value, which leads to a high $V_{OC}$ and FF. An increase in anatase content in photoelectrodes with 20/80 and 40/60 A/R ratios resulted in a decrease in the $R_{rec}$ values, leading, consequently, to the decrease in FF and overall PV performance of PSCs [36]. A PSC sample with the A/R ratio of 60/40, being an exception from this trend, is characterized by efficient electron transfer together with a high $R_{rec}$ value. This leads to optimized charge extraction and low recombination, providing the best PV performance among the PSCs under study.

**Table 4.** Parameters of equivalent circuit obtained by fitting the experimental EIS spectra for PSCs with different A/R ratios.

| Sample | $R_s$ ($\Omega$) | $R_{ct}$ ($\Omega$) | $C_{hf} \cdot 10^8$ (F/cm$^2$) | $R_{rec}$ ($\Omega$) | $C_{lf} \cdot 10^4$ (F/cm$^2$) |
|--------|--------|--------|--------|--------|--------|
| A | 9.7 | 273.0 | 8.6 | 603.0 | 7.8 |
| A/R 80/20 | 7.7 | 326.0 | 7.8 | 700.0 | 6.7 |
| A/R 60/40 | 8.4 | 156.0 | 7.8 | 2400.0 | 15.5 |
| A/R 40/60 | 9.4 | 122.0 | 6.6 | 1150.0 | 18.9 |
| A/R 20/80 | 10.2 | 140.0 | 6.7 | 1700.0 | 12.5 |
| R | 7.6 | 161.0 | 6.8 | 3260.0 | 10.5 |

Obtained EIS results could be attributed to the interplay between two competing electron extraction pathways (from perovskite to anatase and from perovskite to rutile) in mixed-phase $TiO_2$ ETLs. Efficient charge injection from perovskite to anatase is provided by the high surface area of the $TiO_2$ mesoscopic layer and high electron mobility [27]. On the other hand, the anatase phase possesses a larger density of surface defects acting as recombination centers [38], leading to increased recombination at the $TiO_2$/perovskite interface and lower $V_{OC}$ values compared to rutile-based PSCs. The rutile phase in mixed-phase ETLs provides a competing pathway of charge extraction from the perovskite layer, being as effective as the anatase pathway. Energy band alignment at the anatase/rutile/perovskite interfaces deduced from XPS data (Figure 6) shows that the conduction band of rutile is 0.2 eV higher than that of anatase. Thus, the electron transport from rutile to anatase particles is possible in $TiO_2$ photoelectrodes with mixed-phase composition. Moreover, rutile in the mixed-phase ETL could block electrons injected into the anatase phase from recombination at the $TiO_2$/perovskite interface, leading to increased $R_{rec}$ for PSCs based $TiO_2$ layers with high (>40%) rutile content.

Capacitance vs. frequency plot is one of the possible representations of the EIS data [39]. The real part of the complex capacitance $C$ is defined as

$$C'(\omega) = \frac{Z''}{\omega |Z(\omega)|^2}$$

where $\omega = 2\pi f$, $|Z| = \sqrt{(Z')^2 + (Z'')^2}$, $Z'$—real part of impedance, $Z''$—imaginary part of impedance. The capacitance vs. frequency plots for the PSCs with different A/R ratios are presented in Figure 8.

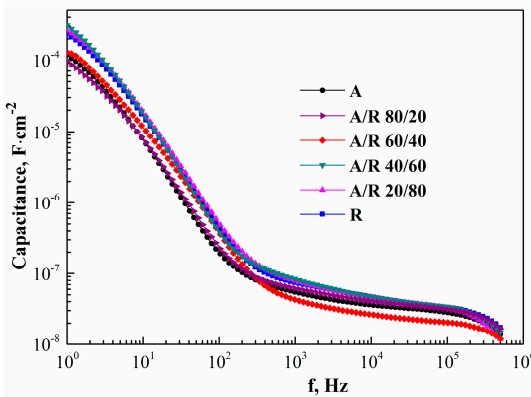

**Figure 8.** Logarithmic plots of capacitance vs. frequency for the investigated PSCs with different A/R ratios.

High frequency capacitance corresponds to dielectric polarization of the perovskite layer [37], while the low frequency capacitance could be attributed to charge accumulation at the perovskite/TiO$_2$ interface under illumination [37,39]. As shown in Figure 8, the bulk capacitance of the perovskite layer is generally independent of the A/R ratio in photoelectrodes. However, higher values of low-frequency capacitance were observed for PSCs with predominant rutile content in photoelectrodes (R, A/R 20/80, A/R 40/60) compared to PSCs with predominant anatase content in photoelectrodes (A, A/R 80/20, A/R 60/40). This observation points to an increased charge accumulation at the perovskite/TiO$_2$ interface for PSCs from the former group. As was shown recently, charge accumulation at the perovskite/TiO$_2$ interface promotes interfacial recombination, which could reduce the extracted photocurrent [39,40]. Thus, the observed decrease in the $J_{SC}$ values on an increase in rutile content (Table 2) is in good agreement with obtained EIS data.

Obtained XPS results together with the EIS data could give a rationale for the observed increase in PCE upon the addition of 40% of rutile. For the anatase-based PSCs, efficient electron extraction is provided by high surface area of the TiO$_2$ mesoscopic layer and large contact area with perovskite. The addition of the rutile phase into TiO$_2$ ETL adds another pathway of charge extraction from the perovskite layer, which could be equally effective as deduced from the EIS data. Specific energy band alignment at the anatase/rutile/perovskite interfaces deduced from the XPS data provides the possibility for the interparticle electron transport from rutile to anatase phase and efficient blocking of electron recombination at the TiO$_2$/perovskite interface, leading to the efficient electron-hole separation in PSCs based on mixed-phase TiO$_2$ photoelectrodes.

Another important aspect that should be taken into account is the difference in electron mobility between the two TiO$_2$ phases. Electron mobility for anatase is higher compared to rutile, as shown both for single TiO$_2$ crystals and nanocrystalline anatase and rutile [38,41,42]. Moreover, the free electrons are deeply trapped at defects in rutile [43], thereby reducing electron mobility and leading to lower electron recombination rate as was reported previously for rutile-based PSCs compared to anatase-based ones [27]. This could account for the high $V_{OC}$ values observed for rutile-based PSCs (Table 2). On the other hand, since electron mobility in rutile is low, charge accumulation at the perovskite/TiO$_2$ interface became dominant for PSCs with high rutile content in photoelectrodes (R, A/R 20/80, A/R 40/60). This leads in turn to the decrease in the $J_{SC}$ values on an increase in rutile content (Table 2). PSCs based on TiO$_2$ ETL with 60/40 A/R ratio were characterized by optimized charge extraction and low level of charge accumulation and recombination at the perovskite/TiO$_2$ interface, providing the best PV performance (PCE = 13.4%) among the fabricated PSCs with various A/R ratios.

Summing up the XPS and EIS data obtained, it could be deduced that for mixed-phase TiO$_2$ photoelectrodes an optimal A/R ratio exists providing optimized charge extraction from perovskite layer, decreased charge accumulation at the perovskite/TiO$_2$ interface and low recombination. Therefore, an increase in PV performance observed for PSCs based on mixed-phase TiO$_2$ photoelectrodes with an

optimized A/R ratio (60/40) was due to balanced processes of electron injection (from perovskite to anatase and rutile) and electron transfer across the more conductive anatase phase.

## 4. Conclusions

In this study a set of $TiO_2$ thin films with varied anatase/rutile mixed phase composition, deposited on conductive glass substrates, was prepared. XRD, optical spectroscopy, SEM and XPS were employed to investigate the structure, morphology and optical characteristics of $TiO_2$ powders and $TiO_2$-based thin films. XPS analysis revealed a cliff conduction band alignment of 0.2 eV for anatase in respect to the rutile. Energy band alignment at the anatase/rutile/perovskite interfaces deduced from the XPS data provides the possibility for the interparticle electron transport from rutile to anatase phase and efficient blocking of electron recombination at the $TiO_2$/perovskite interface, leading to the efficient electron-hole separation in PSCs based on mixed-phase $TiO_2$ photoelectrodes. Obtained EIS results could be attributed to the interplay between two competing electron extraction pathways (from perovskite to anatase and from perovskite to rutile) in mixed-phase $TiO_2$ ETLs. It could be deduced that for mixed-phase $TiO_2$ photoelectrodes an optimal anatase/rutile ratio exists providing optimized charge extraction from the perovskite layer and low recombination at the perovskite/$TiO_2$ interface.

It was shown that the best PCE value of 13.4% was achieved for PSC based on $TiO_2$ mixed-phase photoelectrode with an anatase/rutile ratio of 60/40. According to EIS and XPS analysis, this optimized mixed-phase $TiO_2$ system is characterized by efficient electron transfer and reduced recombination at the perovskite/$TiO_2$ interface. The obtained results provide a simple and effective approach for optimization of the anatase/rutile ratio in mixed-phase $TiO_2$ photoelectrodes and improvement of PV performance in $TiO_2$-based PSCs.

**Supplementary Materials:** The following are available online at http://www.mdpi.com/2071-1050/12/3/788/s1.

**Author Contributions:** Conceptualization, A.N.; methodology, A.N., M.V. and L.L.; formal analysis, A.N., S.K. and N.T.; investigation, A.N., M.V. and S.K.; writing—original draft preparation, A.N.; writing—review & editing, S.K., N.T. and L.L.; supervision, L.L. and O.S.; project administration, O.S. All authors have read and agreed to the published version of the manuscript.

**Funding:** This work was supported by the Russian Foundation for Basic Research (grant No. 19-08-01042), Korea Research Fellowship Program (2019M3E6A1104196) and Brain Pool Program (2019H1D3A2A01062040) through the National Research Foundation funded by the Ministry of Science and ICT of Korea.

**Conflicts of Interest:** The authors declare no conflict of interest.

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
