# Peer review of "Charge Transfer in Mixed-Phase TiO2 Photoelectrodes for Perovskite Solar Cells"

_sustainability, doi:10.3390/su12030788_

Round 1

Reviewer 1 Report

The authors present here a study on the effect of anatase:rutile ration in mesoporous TiO2 for efficient perovskite solar cells. The work is interesting, however, in my opinion, there is a fundamental lack of experiments to support the authors' findings. By changing TiO2, the whole perovskite crystallization changes. Thus, the perovskite ontop is totally different in each case. Having that said, you cannot attribute the observable changes only to the substrate, but the perovskite film itself could be responsible for your results. So, for a proper study, crystallization of perovskite should be studied with XRD, UV-Vis, PL etc. 

Additionally, the thickness of TiO2 is very variable from 200 to 400 nm. This will also change the crystallization of the perovskite ontop. 

This is totally different when compared with dye solar cells when you simply adsorb a dye on TiO2 particles. 

Reviewer 2 Report

I like this paper, it's a good piece of work but I have a couple of major issues with it. The main one is your conclusion. I don't think you provide a strong enough case for it being a two-step charge transfer process. I don't think your paper would be weaker if you took this out, and showing that would be significantly more work.

Optical spectroscopy: you only do reflectance, why? Can you not do absorbance or transmission? I would like to see this in the main paper for at least rutile and anatase, if not also the mixtures. 

Device data: Your IPCE is not clear what it actually is. If you measured short-circuit current please say so.

You say you made a lot of devices, but then you give a simple number for device parameters, I want to know standard error values (or standard deviation, but tell me why it's not standard error), or see histograms. If you are going to present hero cell values tell me why you're doing that. Don't give me three or four significant figures with no justification.

I think experimental details are probably missing for how you actually did your experiments, you just say what equipment you use for the most-part. Please rectify this (especially XPS, EIS, IPCE and XRD)

The impedance spectroscopy is confusingly presented

The English is mostly comprehensible, except in the introduction, which needs a bit of work. Other than one non-Latin character that got left behind (line 266, use and or &) the main text is well translated and transliterated.

Detailed feedback:

Line 28 - what environmental protection? (check ref 1,2) this is poorly defined

32 processes

33 create a platform

35 and et. -> etc.

36 competitive application field doesn’t make sense… level playing field?

40 PSCs are based on ABX3 formula, A=methylammonium, formamidinium, Cs; B=Pb, Sn

44 this is only one limitation

53 than the rod-like structure

71 which reference is scanlon et al. not super clear

92 IPCE is also sometimes known as IQE or EQE, which I prefer, especially in this context as that’s what your instrument is called, and since you’re presumably measuring this in short-circuit it’s not really internal power conversion efficiency. Please either address this concern or make this IQE or EQE (not sure which you actually did)

143 using a DRON-3M X-ray

Figure 1: tell me more about how you did this XRD measurement. Angle step size, integration time etc. Also why do I care, you mixed some ground powders and the peak height ratios had the ratio of the precursors (without errors, please add) - so what? I would be more interested in the XRD of ETL layers made from these powders - does that match up?

Figure 2: maybe put ratios on images so they’re easier to read, is the scale the same in all the figures? If so, say so, also make the scale bar easier to see. I’m not actually sure that looking at these images without statistics is particularly illuminating. I would rather see the statistics. Do you see two distributions of particle sizes, or one centered on the mean size? That’s what it looks like, but hard to say definitively. 

164-171: I would like to see optical spectra in the main text, as well as see the spectra of the mixed materials (at least in the SI for the mixtures).

197-198 this sentence isn’t really supported. Why would this non-uniformity lower their performance? What are your statistics on the VOCs? That would be very informative. It seems like it’s really a balance between good extraction for anatase and low voltage barrier for rutile. It’s a shame your PCEs are so low, because it would be cool to know if this trend is the same for high performance cells

Figure 3: I would like to see zoomed-in images of open-circuit voltages, as these are hard to distinguish. I think this would be a lot more informative than your ‘IPCE’ plot which just shows the same trend as your JSC.

Figure 4S and table 1S need more than just ‘average’ Are these the mean values? What are the error bars? Standard deviation or standard error (which is appropriate and why?) are all 30 devices functioning? I would like to see histograms and more detailed stats here, because I think this is really the heart of your story.

Figure 4 and 5 should really be part of the same figure, side by side would be nice.

232: EF looks weird. Please put the F in a subscript and define this earlier, or just say Fermi energy.

232/235 - what is this optical analysis? Are you just talking about literature values, or did you do this? THis is the worst part of this story so far!.

249: Please use subscripts

266: what is this symbol: ‘и’? Please transliterate this.

274-294: please make your definition practice consistent here. I like the convention of, eg: series resistance [RS] as this is easy to locate in the article and distinguishes the square brackets from round brackets used for other parentheses.

I think 260-316 could be made a lot easier to read if you restructure a bit. I like that you have your equivalent circuit model described here because it’s easier to refer to, but I would maybe give a small intro to this section, give the model and the spectra in one figure with the data table below, and then talk about the data in that table next so that the reader can refer to it easily.

Figure 8: you change your lable from Cg to CPEg here, which is confusing. Don’t do this. Also use parallel lines for a capacitor, not the concatenated Vs here.

281: you change your A/R convention here ‘A/Ru’

291: is it an assumption? Probably more like a conclusion or interpretation

298: this formatting is really hard to read. You need to fix this.

305: please make sure you don’t have line breaks between numbers and units or figures/tables and their number. This is also hard to read.

Table 4 & 305-316: what are your fitted capacitances? It would be much easier to understand your arguments if I had these numbers. If you are going to show your capacitance/frequency plot please give an equation for your equivalent circuit’s overall capacitance’s frequency dependence. This is hard to follow.

320: You mean rationale, not rational. Rationale is a noun that is an argument in favour of something. Rational is an adjective.

325-329: I don’t think you satisfactorily demonstrate two-step charge transfer. Either provide more evidence for this than an assertion, suggest it tentatively, or remove it. What are the two steps? Because as far as I can see the rutile has a lower voltage barrier, and the anatase has a higher surface area. Would having the less smooth interface give a higher effective surface area for the rutile phase, lowering the contact resistance between the rutile and the perovskite? I just don’t think you can conclusively justify that this is a two-step process. 

332-336: these sentences are repetitive and could be combined for greater clarity/

336: VOC, Voc, VOC or Voc. Pick one

357-264: I think you should remove these lines from the conclusion. I don’t need the experimental details in the conclusion, and I don’t believe that you have conclusively demonstrated the two-step process. If you could get rutile particles the same size as the anatase and show the same trends, maybe that would help

Reviewer 3 Report

Attached please find my comments.

Round 2

Reviewer 1 Report

I carefully read the authors' feedback. I would like to see also a comment on how the different thickness of TiO2 (depending on the preparation conditions and anatase to rutile ratio) affects (mainly) the Jsc of the devices. 200 or 400 nm could play a severe role here for proper electron extraction. 

Reviewer 3 Report

The manuscript revised well. The reviewer thought it could be accepted to publish.

Author Response

We would like to thank the Reviewer for careful reading of our manuscript and for all constructive remarks.